# Different In Silico Approaches Using Heterocyclic Derivatives against the Binding between Different Lineages of SARS-CoV-2 and ACE2

**DOI:** 10.3390/molecules28093908

**Published:** 2023-05-05

**Authors:** Federica Sipala, Gianfranco Cavallaro, Giuseppe Forte, Cristina Satriano, Alessandro Giuffrida, Aurore Fraix, Angelo Spadaro, Salvatore Petralia, Carmela Bonaccorso, Cosimo Gianluca Fortuna, Simone Ronsisvalle

**Affiliations:** 1Department of Drug and Health Sciences, University of Catania, Viale A. Doria 6, 95125 Catania, Italy; 2Department of Chemical Science, University of Catania, Viale A. Doria 6, 95125 Catania, Italy

**Keywords:** SARS-CoV-2, molecular modeling, heterocyclic derivatives

## Abstract

Over the last few years, the study of the SARS-CoV-2 spike protein and its mutations has become essential in understanding how it interacts with human host receptors. Since the crystallized structure of the spike protein bound to the angiotensin-converting enzyme 2 (ACE2) receptor was released (PDB code 6M0J), in silico studies have been performed to understand the interactions between these two proteins. Specifically, in this study, heterocyclic compounds with different chemical characteristics were examined to highlight the possibility of interaction with the spike protein and the disruption of the interaction between ACE2 and the spike protein. Our results showed that these compounds interacted with the spike protein and interposed in the interaction zone with ACE2. Although further studies are needed, this work points to these heterocyclic push–pull compounds as possible agents capable of interacting with the spike protein, with the potential for the inhibition of spike protein–ACE2 binding.

## 1. Introduction

Coronaviruses (CoVs) are a large, highly diverse, single-stranded, positive-sense group of RNA viruses. They belong to the subfamily *Coronavirinae* of the family *Coronaviridae* [1]. In the last three years, a new type of coronavirus called SARS-CoV-2 has caused several serious human diseases involving the respiratory, enteric, hepatic, and neurological systems [2,3]. The structure of SARS-CoV-2 is summarized as follows: the nucleocapsid protein (N) forms a capsid outside of the genome, which is further packed by an envelope associated with three structural proteins: membrane (M), spike (S), and envelope (E) proteins, and another 16 nonstructural proteins (NSP1-NSP16). Each of these NSPs plays a different role in the attachment, penetration, replication, and release of the virus [4]. The spike protein is a trimeric glycoprotein expressed on the surface of the coronavirus involved in the entry of the virus into the host cell. The trimeric structure is characterized by a large ectodomain, a single-pass transmembrane anchor, and a short intracellular tail. The ectodomain contains two cleavage sites that are targets of transmembrane serine protease 2 (TMPRSS2). Protease cleavage is necessary for viral entry into the cells. During this phase, the spike protein is cleaved into a receptor-binding S1 subunit and membrane-fusing S2 subunit [5]. S1 binds to a receptor on the host cell surface for viral attack, and S2 fuses host and viral membranes, allowing viral genomes to enter host cells [6]. Receptor binding and membrane fusion are the initial and critical steps in the coronavirus infection cycle. Specifically, in the S1 portion of the spike protein, two main S1 domains have been identified: the N-terminal domain (S1-NTD) and the C-terminal domain (S1-CTD). Specifically, the S1-CTD domain is responsible for the recognition of angiotensin-converting enzyme 2 (ACE2), aminopeptidase N (APN), and dipeptidyl peptidase 4 (DPP4). Therefore, it was identified as the receptor-binding domain (RBDs) [7,8,9,10]. Due to their small genome and replication mechanism, viruses are easily adapted by mutating their genome, which is reflected in structural mutations that could facilitate their interaction with the host organism [11]. In 2003, Ruan Y. et al. clearly expressed that these mutations in the S1 region could be a starting point for the virus to survive at the human immunological advance [12]. The rapid increase in COVID-19 cases that occurred during the pandemic years has been attributed to numerous mutations in the viral structure, some of which have been shown to be critical for improved viral transmission or immune escape [13]. Relevant mutations, such as those that involve N501Y, expressed in the spike protein, promote an increased affinity of the virus for ACE2, which leads to an improvement of the viral transmission and, in conclusion, enhanced infectivity. Another example is the N439K mutation that allows the virus to elude antibody-mediated immunity [14,15,16]. The D614G mutation present in the spike protein near the RBD is one of the most prevalent in the different viral variants. This probably occurs because mutating an aspartic amino acid into a glycine residue could improve the flexibility of the protein and enhance the binding to ACE2, thereby increasing virulence [17]. The study of the structure of viral proteins is indispensable for understanding their function and using this knowledge to study new ligands. For this purpose, several molecular modeling studies have investigated the structural basis for viral recognition. Some of these, conducted on SARS-CoV-2, have focused on the spike protein, which mediates entry into the host cell by interacting with the cell surface receptor. The molecular dynamics technique, which can predict the temporal evolution of molecular systems, was applied to the spike protein–ACE2 complex to simulate their interactions and derive useful information for vaccine and drug development [18,19]. These studies would not have been possible without the crystallographic structure of the spike protein complex bound to the ACE receptor (protein data bank code 6M0J), which allowed us to understand the interaction between the RBD (residues Arg319–Phe541) of SARS-CoV-2 and ACE2 at a higher resolution [20]. The results of virtual screening studies to investigate the affinity and binding mode of some small molecules, some of which are derived from fatty acids with the SARS-CoV-2 spike protein, showed that it is indeed possible that some small molecules can bind to the spike protein to prevent its interaction with human ACE2 [21,22]. This study aimed to further investigate the areas of the spike protein involved in interaction with the host organism, and, in particular, to identify possible interactions between small molecules as polycyclic pyrimidine compounds (Figure 1) and the spike protein of the most relevant variants of SARS-CoV-2 using molecular modeling. In particular, the selected variants were Alpha (B.1.1.7), Beta (B.1.351), Gamma (P.1), Delta (B.1.617.2), and Omicron (B.1.1.529). The compounds subjected to virtual screening were push–pull heterocyclic compounds, with a different electro-attractor ring, chosen for their structural varieties and different biological activities (Figure 1). Previous studies have demonstrated that biological activity varies completely by varying the presence of a relevant electro-attractor component, from a quinoline ring to a pyridine and imidazole ring. This is relevant in this case where the interaction occurs between two different biological entities, such as the spike protein S1 subunit and a protein transmembrane receptor [23]. These compounds have shown to be excellent DNA intercalators due to these chemical characteristics [24]. These characteristics, as well as their planarity, could disrupt the interaction between the SARS-CoV-2 spike protein S1 subunit and the ACE2 receptor, and using them as lead compounds will result in further structural modifications to improve the selectivity toward the spike protein–ACE2 complex.

## 2. Results

### 2.1. Binding Site Identification and Molecular Docking Studies

The first screening was performed on the viral wild-type and FLAP identified three pockets (Figure 2A). For convenience, the pockets were labeled as 1, 2, and 3, and the amino acids that characterize them are listed in Table 1. For pocket 1, the compound with the highest Glob-Sum score (1.583), according to FLAP, was BCC2, as listed in Table 2. Pocket 1 is located between the α-helix of the ACE2 receptor and the β-sheets of the spike protein (Figure 2A, blue pocket). Overall, BCC2 strongly interacted with Trp476, Pro470, Pro469, and Tyr436. From the 2D representation (Figure 3B), the software highlighted the most involved interaction areas, the donor and acceptor of hydrogen bonds (red and blue, respectively), and an area of interest in hydrophobic bonds (DRY—green area) involving the quinoline group. In particular, Trp476 and Pro470 exhibited interesting π–π and CH–π interactions. Pocket 2 is located lateral to the crystalline structure and is smaller than pocket 1 (Figure 2A, green pocket). BCC1 (Glob-Sum 1.076) and BCC2 (Glob-Sum 1.037) had similar scores and interesting positions. In both cases, the bicyclic system (4-(pyrimidin-5-yl)-phenyl) fitted into the pocket as if the two compounds were intercalators (Figure 3C). BCC3 had a low Glob-Sum score (0.358) and a different pose compared to the first two compounds. The areas of pocket 2 most involved in the interaction with BCC1 were the hydrogen bond donor and acceptor characters (Figure 3D). Residues Pro462 and Phe460 interacted via supramolecular π–π and CH–π interactions with the bicyclic system. Pocket 2.5 is an “enlargement” of pocket 2 (Figure 2A, yellow pocket). Its larger size allows a new laying of BCC3 (Figure 3E) and an increase in the areas of the pocket affected by hydrophobic interactions. The amino acid residues that have the most interactions with the compound under examination are Gly446 and Tyr408, which bind via weak interactions with the disubstituted imidazole group with two methyl groups and the bicyclic system common to all three BCC compounds (Figure 3F).

For the **Alpha** variant, FLAP identified three pockets (Figure 2B). These are located in three different areas of the S1-CTD portion. Pocket 1 is located in an upper area of the spike protein (Figure 2B, blue pocket), pocket 2 is located in a region between the β-sheets of the spike and the α-helix of the ACE2 receptor (Figure 2B, yellow pocket), and pocket 3 is laterally located to the β-sheets of the spike (Figure 2B, red pocket). The last two pockets are shared between spike protein and ACE2 receptor. Pocket 1 is the smallest of the three pockets of the Alpha variant, located in an area where the ACE2 receptor is absent, and is located in an upper pocket between the β-sheets and α-helix of the spike protein (S1-CTD). BCC3 is the compound with the highest Glob-Sum score (1.801). From the 2D representation, it can be seen that the functional group most involved in the interactions is imidazole disubstituted with two methyl groups, which interact most with residues Tyr369 and Tyr365, and, as shown in Figure 4B, are hydrophobic and hydrogen bond acceptors. From the 3D pose, it can be observed that the compound acts as an interlayer for pocket 1 (Figure 4A). Pocket 2 is located in the central area of the crystallized structure and has the largest extension of the three. This pocket has many affinities with the central pocket of the wild-type protein. The molecule with a high Glob-Sum score is BCC3 (1.601). Pocket 2 has a greater depth and surface area than pocket 1, and the pose showed greater interaction with the portion related to the ACE2 receptor (Figure 4C), while the 2D representation (Figure 4D) shows how the areas of the hydrogen bond donor character and hydrogen bond acceptors (red) have increased (blue) and areas of hydrophobic character (DRY) have decreased. In this case, imidazole is the portion with the greatest interactions, specifically with residues Tyr478 and Phe72, the latter being related to the ACE2 receptor. pocket 3 is very small compared to the other two and shares some amino acid residues with the ACE2 receptor. For the reasons described above, this pocket had no affinity for the three compounds under examination.

For the **Beta** variant, the software identified three pockets: (Figure 2C); pocket 1 is in the central area among the β-sheets (Figure 2C, blue pocket), pocket 2 (Figure 2C, yellow pocket) is at the top with a much smaller surface, and pocket 3 (Figure 2C, red pocket) is at the bottom that is shared between the ACE2 receptor and S1-CTD. Pocket 1 has a larger surface area than the other two pockets and has a small portion shared with the ACE2 receptor. BCC1 had a higher Glob-Sum score (1.393) and a lower DRY score (0.488) than the others. From the 3D pose, it was observed that BCC1 was entirely incorporated into the pocket (Figure 5A). The major interactions have a hydrogen bond donor/acceptor nature. Pocket 2 has many similarities with pocket 1 of the Alpha variant because, despite being slightly more laterally displaced, it still has the same surface and size. This was also observed between the α-helix and β-sheets of the spike. BCC1 had the highest Glob-Sum score (1.843). The 3D pose shows that the molecule could act as an intercalator (Figure 5C). Figure 2D confirms the score, with the largest areas related to hydrophobic interactions and hydrogen bond acceptors. Tyr365 interacts with the ethylene bridge and the first ring of the bicyclic system (Figure 5D) with π–π, CH–π interactions. Pocket 3 has a small surface, and its position is shared between the ACE2 receptor and the spike protein. BCC2 showed the highest Glob-Sum score (1.594). The 3D representation confirmed the intercalating interaction (Figure 5E), and the 2D image showed that the most affected areas are those that provide hydrogen bond acceptor interactions. As shown in the figure, the amino acid residue that interacts most with the compound is Tyr478. The residue interacts simultaneously with both the quinolinium group and the bicyclic group, including the ethylene bridge (Figure 5F). The interaction with this residue, and with Trp479, is also repeated with the other two compounds.

The **Delta** variant has more pockets than previous variants (Figure 2D). FLAP identified four pockets: specifically, pocket 1 in the upper part of the structure (Figure 2D, blue pocket), while the other three are in the lower part. They also contain amino acid residues that belong to the ACE2 receptor. Pockets 2 (Figure 2D, yellow pocket) and 3 (Figure 2D, red pocket) seem to almost overlap and pocket 3 is more secluded just above the ACE2 receptor α-helix. In contrast, pocket 4 is located in the central area of the structure (Figure 2D, green pocket). Pocket 1 of the Delta variant showed several affinities from pockets 2 and 1 of the Beta and Alpha variants (location, surface area, and size). FLAP gave very high interaction scores for all three compounds, the most interesting being BCC2 with a high Glob-Sum score (2.448) and a high DRY score (1.782). The compound appeared to act as an intercalator with pocket 1 (Figure 6A). From the 2D representation, the areas most involved in the bond were hydrophobic in nature and concerned the quinolinium group, leaving a cycle of the bicyclic system outside of the pocket. The 2D representation highlighted interesting π–π interactions between the Tyr369 residue and the quinolinium group. Areas of hydrogen bond interaction were also highlighted (Figure 6B). Pocket 2 exhibited low interaction values (Glob-Sum) with all compounds and no interaction with BCC3. BCC2 had the highest Glob-Sum score (1.458) and showed a very interesting pose. In fact, the compound seems to slip into the pocket as if it were an intercalator. The 3D pose showed that a ring of the bicyclic group stays out of the pocket (Figure 6C). A 2D representation showed that the hydrogen bond donor/acceptor areas are greater, and the areas related to hydrophobic interactions are reduced (Figure 6D). The residue that interacted with the quinolinium system is Phe464, whereas the residue that interacted the most with the compound is Val461 (Figure 6C). Pocket 3 has many similarities with the previous pocket (size and surface) but is located in a lower area involving the ACE2 receptor. According to FLAP, the only compound that interacts with pocket 3 is BCC3 (1.018). The interactions mainly occurred between Leu29 and the quinolinium group. Furthermore, a reduction in DRY interactions was shown (Figure 6F). Pocket 4 is located in the central area of the spike protein, between spike β-sheets, and is shared with the ACE2 receptor. It did not have a large surface area and is the smallest of the four pockets of the Delta variant. Unlike the previous two pockets, this pocket showed significant interactions with the three compounds. The compound with the highest interaction score was BCC3 (Glob-Sum 1.362), despite its very low hydrogen bond acceptor score (0.042) and hydrophobic interaction (0.590). The compound exhibited a high steric interaction score (H 0.799) with the pocket. The 3D pose showed that the pyrimidine group remains outside the pocket (Figure 6G) and how major DRY and O interactions occur with the imidazole group. Two amino acid residues, Tyr443 and Val407, interact with the bicyclic system (Figure 6H).

The **Gamma** variant exhibited six interaction sites. The pockets, except for the 3.5, are not large but simultaneously touch different areas of the structure. Pockets 1 (Figure 2E, blue pocket), 2 (Figure 2E, purple pocket), and 5 (Figure 2E, dark blue pocket) consist of amino acids belonging exclusively to the spike protein, whereas the others are shared with the ACE2 receptor and are located at the interface of the complex S1AS. Furthermore, the Gamma variant exhibits the most interesting molecular dynamics. Pocket 3.5 (Figure 2E, yellow pocket) is an “enlargement” of pocket 3 (Figure 2E, red pocket) and is located in the central area of the target. Pocket 1 did not have a large surface area and is located in the upper area of the S1-CTD portion, specifically above the spike β-sheets. Unlike the other variants, the α-helix motif that accompanies the pockets in this area is not observed. For this pocket, the Glob-Sum scores were similar but BCC2 had the highest score (1.772). BCC2 is perpendicular to pocket 1. Only the quinolinium group and part of the bicycle group were inside the pocket (Figure 7A). The affected areas were predominantly hydrophobic (DRY) and hydrogen bond acceptors (O). Phe338 is the residue that interacts more with BCC2 (Figure 7B). Pocket 2 has dimensions comparable to the previous pocket and is located in the lateral area of the spike protein. It did not interact with any of the three compounds. Pocket 3 is located at the center of the protein structure, has dimensions comparable to previous ones, and does not have high interaction values with the three compounds. It is shared by the ACE2 receptor and is located between the spike β-sheets. BCC2 had the highest Glob-Sum score (1.238) for this pocket. Unlike pocket 1, the bicyclic system is inside the pocket, unlike the quinolinium group (Figure 7C). The residues that mostly interact were Tyr489 and Tyr505. The quinolinium group strongly interacted with the Phe486 residue (Figure 7D). Pocket 3.5 is an extension of the last pocket, and has been searched manually to consider the same position as the previous pocket, but its size was increased to determine whether interactions can increase. The interaction values increased considerably owing to the larger pocket size. However, these compounds do not act as intercalators but are completely incorporated into this pocket (Figure 7E). By increasing the size of the pocket, the interactions of a hydrophobic nature (DRY) significantly increase. The amino acid with the greatest interaction was Trp479. This interaction was more pronounced in bicyclic systems (Figure 7F). Pocket 4 is located low in the target protein between the α-helix of the ACE2 receptor and below the β-sheets of the spike protein (Figure 2E, green pocket). It has small dimensions and contains amino acid residues from both spike and ACE2 receptors. Moderate interaction values were observed in this pocket. BCC1 interacted the most (Glob-Sum 1.363), even if the significant interactions are with the ACE2 receptor and not with the spike protein. The spike residue that interacted the most with BCC1 and with the other two compounds was Trp479 (Figure 7H). This molecule interacts with the bicyclic systems (Figure 7G). Pocket 5 has small dimensions; it is located lateral to the spike β-sheets in an area where the α-helix exists in the previous variants. As for pocket 1, pocket 5 showed good interaction scores. Specifically, there were good DRY (1.034), O (0.086), and H (0.710) scores. Compound BCC2 is placed horizontally with respect to the pocket and places the pyrimidine group outside the pocket (Figure 7I). The group that interacts directly with the pocket is always the quinolinium group, even if the compound is placed in such a way that it can rotate inside the pocket. The areas most involved are only the DRY and O (Figure 7J).

For the **Omicron** variant, FLAP identified four different pockets. The peculiarity of these pockets is that most of them are concentrated in the central area of the spike protein; in fact, pockets 1 (Figure 2F, blue pocket), 2 (Figure 2F, yellow pocket), and 3 (Figure 2E, red pocket) are shared with the ACE2 receptor. The only pocket found exclusively in the spike protein is pocket 4 (Figure 2F, green pocket), which is specifically located above the β-sheets of the spike and laterally to the α-helix. Pocket 1 has the smallest area and it is located in the region that interacts with the ACE2 receptor. These three compounds have low Glob-Sum scores. BCC1, which is the highest Glob-Sum scoring compound (1.132), is fitted vertically inside pocket 1 (Figure 8A). BCC1 exhibited excellent hydrogen bond (O 0.129) and steric character (H 0.634) interaction scores. From the 2D pose, it can be seen that the bicyclic system is inside the pocket and that the amino acid residue that gives the greatest interactions was Pro473. Interactions with this residue involved both parts of the bicyclic system and the ethylene bridge (Figure 8B). Pocket 2 is identified by FLAP above the α-helix of the ACE2 receptor and laterally to the β-sheets motif of the spike. This pocket has the largest dimensions. The compound that interacts most is BCC1 (Glob-Sum 1.796), even if all three compounds have high and similar Glob-Sum scores. For this compound, the pyridine group remained partially outside the pocket (Figure 8C). BCC1 exhibited high scores for DRY (1.017) and O (0.346), whereas the steric interactions were comparable in all three compounds. The 2D pose showed a large hydrogen bond donor interaction area (N1), a small hydrophobic bond area (DRY), and a hydrogen bond acceptor area (O). The residue that presented the strongest interactions was Tyr495, which interacted with the pyridinium group through π–π interactions (Figure 8D). Pocket 3 is located in a region common to the other variants, centrally between the β-sheets of the spike and the α-helix of the ACE2 receptor. The pocket has a discrete surface such that it is shared with the ACE2 receptor. FLAP considered the BCC2 compound as having the most interaction because it has the highest Glob-Sum score (1.697). This is mainly due to the scores related to DRY (0.875) and H (0.814). The 3D pose shows how this compound “puts beside” pocket 3 (Figure 8E). From the 2D pose, Trp479 was the most involved residue (Figure 8F). Pocket 4 is located at the top of the crystallized structure, which is also common to other variants. The pocket involves the spike protein β-sheets motif (Figure 8G). For this pocket, scores are comparable, with high O and H scores. The compound that interacts the most was BCC3 (Glob-Sum 1.535). The amino acid residue that interacts the most was Tyr365. This interacted with the ethylene bridge and the first ring of the bicyclic system and with the methyl group of the pyridine ring. These interactions are π–π and CH–π (Figure 8H).

**Figure 8 molecules-28-03908-f008:**
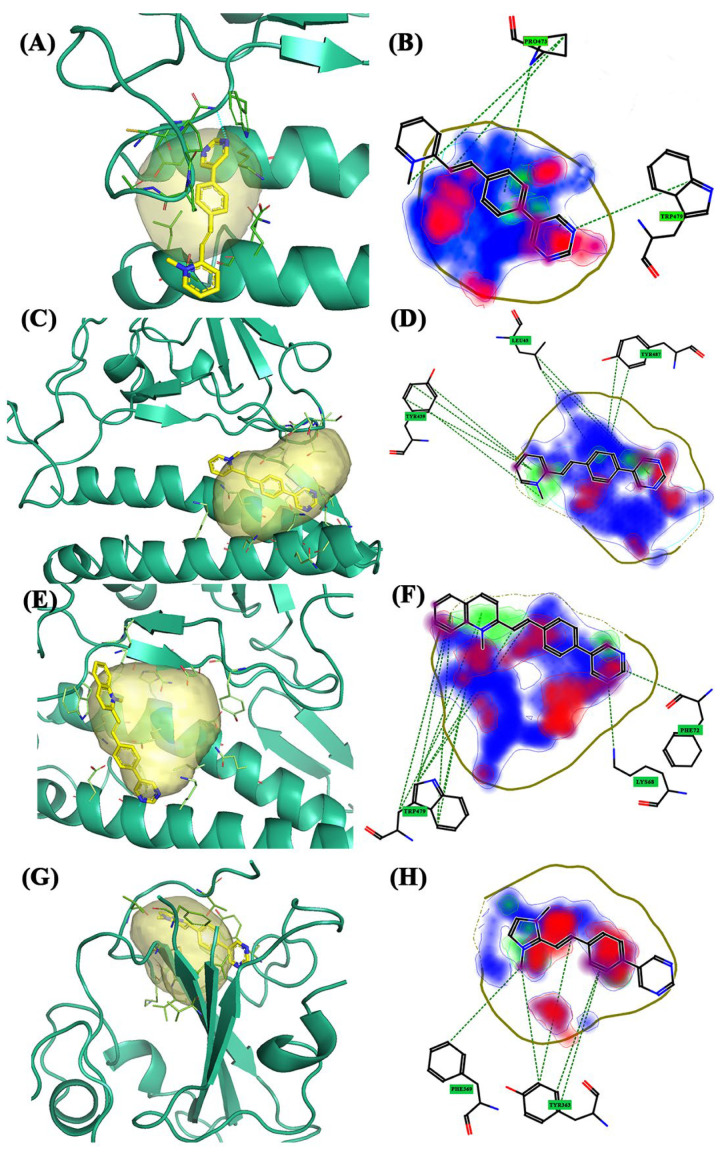
The 3D and 2D docking Omicron variant: (**A**) 3D pocket 1 with BCC1; (**B**) 2D pocket 1 with BCC1; (**C**) 3D pocket 2 with BCC1; (**D**) 2D pocket 2 with BCC1; (**E**) 3D pocket 3 with BCC2; (**F**) 2D pocket 3 with BCC2; (**G**) 3D pocket 4 with BCC3; and (**H**) 2D pocket 4 with BCC3. Water and the accessory parts of the spike protein and ACE2 receptor were omitted for clarity. The text in the green square of the image describes the amino acids involved in interaction with the ligand.

**Table 1 molecules-28-03908-t001:** Summary of all pockets detected in SARS-CoV-2 wild-type and variants. Each pocket is characterized by the amino acids presented in the table.

Variant	Pocket	Amino Acid ACE2	Amino Acid Spike Protein
Wild-Type	Pocket 1	Lys31, His34, Glu35, Ala36, Asp38, and Leu39	Tyr436, Tyr442, Pro469, Pro470, Cys474, Tyr475, Trp476, Leu478, Asn479, and Asp480
	Pocket 2	Ser19, Glu23, Lys26, Thr27, and Asp30	Tyr408, Tyr442, Leu443, Arg444, His445, Phe460, Ser461, and Pro462
	Pocket 2.5	Glu23, Thr27, Asp30, and His34	Gly403, Val404, Ile405, Asp407, Tyr408, Tyr440, Tyr442. Leu443, Arg444, His445, Lys447, Phe460, and Pro462
Alpha	Pocket 1		Tyr359, Leu362, Tyr363, Phe371, Ser377, Cys422, Leu424, and Leu502
	Pocket 2	Phe28, Lys31, Glu35, Asp38, Leu39, Gln42, Lys68, Phe72, and Gln76	Tyr439, Tyr445, Tyr478, Trp479, Leu481, Asn482, and Asp483
	Pocket 3	Leu45, Trp48, Asn49, and Arg103	Thr434, Ser435, Thr436, Tyr492, Thr493, and Thr494
Beta	Pocket 1	Asn33, His34, Glu35, Glu37, Asp38, and Lys99	Lys393, Asp395, Asp396, Arg398, Gln399, Asn407, Ile408, Tyr443, Asn482, Asp483, Tyr484, and Tyr494
	Pocket 2		Cys330, Pro331, Phe332, Ala357, Tyr359, Val361, and Leu362
	Pocket 3	Lys36, Glu75, and Leu79	Lys474, Leu475, Asn476, Cys477, Tyr478, and Trp479
Delta	Pocket 1		Ala357, Tyr359, Ser360, Tyr363, Phe371, Cys373, Thr379, Lys380, Leu381, Asn382, Val385, Cys422, Leu424, Leu502, and Phe504
	Pocket 2		Arg444, Leu446, Arg447, His448, Asp457, Ser459, Asn460, Val461, Pro462, Phe463, Ser464, Gly467, Lys468, Pro469, and Pro480
	Pocket 3	Gln24, Lys26, Thr27, and Asp30	Tyr411, Leu446, Arg447, His448, Gly449, Phe463, and Pro465
	Pocket 4	Asn33, His34, and Glu37	Lys393, Asp395, Asp396, Gln399, Val407, Ile408, and Tyr443
Gamma	Pocket 1		Pro331, Phe332, Val335, Lys350, Ile352, Leu362, Ala387, and Val500
	Pocket 2		Ile400, Ala401, Pro402, Gly403, Asn407, Ala409, Asp410, Lys414, Leu415, and Asp417
	Pocket 3	His34, Glu35, Glu37, Asp38, and Lys99	Lys393, Tyr443, Asp483, Phe486, and Tyr494
	Pocket 3.5	Phe28, Lys31, His34, Glu35, Asp38, Phe72, Gln76, and Lys99	Tyr345, Lys393, Asp396, Tyr439, Asn440, Lys442, Tyr443, Tyr445, Asn460, Tyr478, Trp479, Pro480, Leu481, Asn482, Asp483, and Tyr484
	Pocket 4	Phe28, Lys31, His34, Glu35, Asp38, Phe72, Gln76, and Leu79	Tyr445, Pro473, Tyr478, Trp479, Leu481, and Asn482
	Pocket 5		Phe336, Asn337, Ala338, Phe348, Trp426, Asn427, Thr428, Arg429, Asn430, Ile431, and Arg498
Omicron	Pocket 1	Lys31, Glu75, Gln76, Thr78, and Leu79	Pro472, Pro473, Ala474, and Trp479
	Pocket 2	Gln42, Leu45, Ala46, Asn49, Asn58, Gln60, Asn61, Asn64, Ala65, and Lys68	Arg103, Ser435, Thr496, Gly497, Tyr487, Thr488, and Thr489
	Pocket 3	Lys31, His34, Glu35, Asp38, Leu39, Gln42, Lys68, Phe72, and Glu75	Tyr439, Tyr445, Trp479, Leu481, Asn482, and Asp483
	Pocket 4		Tyr363, Phe369, Asn371, Lys372, Cys373, Val376, Thr379, Gly421, Cys422, Val423, Leu502, and Phe503

**Table 2 molecules-28-03908-t002:** Summary of FLAP’s score for all pockets detected in SARS-CoV-2 wild-type and variants.

Variant	Pocket	Ligand	GLOB-SUM	GLOB-PROD	Distance	DRY	H	O
Wild-type	Pocket 1	BCC2	1.583	0.504	12.719	0.735	0.825	0.102
	Pocket 2	BCC1	1.076	0.477	13.753	0.334	0.561	0.258
	Pocket 2.5	BCC3	1.515	0.591	12.079	0.766	0.945	0.182
*Alpha*	Pocket 1	BCC3	1.801	0.556	11.731	1.090	0.758	0.204
	Pocket 2	BCC3	1.601	0.613	11.754	0.794	0.978	0.159
*Beta*	Pocket 1	BCC1	1.393	0.529	13.057	0.488	0.868	0.074
	Pocket 2	BCC1	1.843	0.511	12.249	1.270	0.595	0.121
	Pocket 3	BCC2	1.594	0.465	12.545	0.895	0.711	0.192
*Delta*	Pocket 1	BCC2	2.448	0.618	10.406	1.782	0.790	0.211
	Pocket 2	BCC2	1.458	0.254	13.385	0.838	0.621	0.032
	Pocket 3	BCC3	1.018	0.434	14.206	0.248	0.686	0.083
	Pocket 4	BCC3	1.362	0.371	13.562	0.590	0.799	0.042
*Gamma*	Pocket 1	BCC2	1.772	0.460	12.756	1.218	0.563	0.068
	Pocket 3	BCC2	1.238	0	14.837	0.722	0.516	0
	Pocket 3.5	BCC2	2.213	0.665	10.196	1.340	0.964	0.240
	Pocket 4	BCC1	1.363	0.433	13.159	0.723	0.685	0.136
	Pocket 5	BCC2	1.640	0.486	12.615	1.034	0.710	0.086
*Omicron*	Pocket 1	BCC1	1.132	0.479	13.589	0.512	0.634	0.129
	Pocket 2	BCC1	1.796	0.633	10.882	1.017	0.965	0.346
	Pocket 3	BCC2	1.697	0.600	12.015	0.875	0.814	0.190
	Pocket 4	BCC3	1.535	0.413	12.913	0.794	0.775	0.114

In conclusion, as explained earlier, with SBVS function, common hotspots (molecular probes) searched between potential active sites of the protein and ligands so that any mutation in the amino acid sequence could change the supramolecular interactions. An example is the redundant pocket in all variants, the central pocket located at the interface between spike and ACE2. FLAP identified this pocket in all variants but with differences in size, area, and location. These differences could be caused by mutations in the amino acid sequence that, as mentioned above, affect the interaction scores with the three compounds. For example, if we consider the Alpha variant and the corresponding central pocket (pocket 2), we see that compound BCC1 has a Glob-Sum score of 1.576, whereas the same compound but in the central pocket of the Beta variant (pocket 1) has a lower Glob-Sum value of 1.393. In detail, the two pockets have different hydrophobic interaction (DRY) scores: in the case of the Alpha variant, there is a score of 0.938, while for the Beta variant, it is 0.488. Another example is pocket 3 of the Gamma variant, which is still in the same region as the previous ones but has a smaller size and surface area, resulting in very low scores. BCC1 in this pocket has a Glob-Sum score of 0.887 and a DRY of 0.443. Therefore, this could be an important clue to how mutations can change supramolecular interactions between compounds and pockets.

### 2.2. Molecular Dynamics Studies

Molecular dynamics (MD) results were first evaluated for all compounds in all pockets found in the wild-type virus and the different variants listed above. The results of the most representative compounds and pockets for this project are examined below.

MD studies for the Beta variant, starting from the FLAP 2.2.2 software-identified pockets, yielded the most interesting results for pocket 3, most likely because of its positioning at the interface of the complex S1AS. Among the three compounds, BCC1 and BCC2 showed the best results, in agreement with the docking assessments. The BCC1 remained in the pocket for approximately 12 ns. The benzene ring of the bicyclic system of BCC1 established π–π interactions with the phenolic ring of Tyr478 from the onset of the simulation until approximately 10 ns. The interaction with the ligand was also enhanced by the hydrogen bond established between the indole ring of Trp479 and the pyrimidine ring of the ligand, up to approximately 12 ns. In addition, BCC1 also established interaction with several amino acids of ACE2, in particular, the pyridinium ring interacted through hydrogen bonds with Glu75 from 4 to 10 ns, and with Thr78 irregularly up to 12 ns. A conformational change in the loop from amino acid Val461 to Leu481 was observed (Figure 9), leading to the pulling out of the ligand. The quinolinium salt, present in BCC2, gives more rigidity and, therefore, less rotatability to the compound. The same conformational change to the loop from Val461 to Leu481 was observed, which led to a loss in the π–π interaction of BCC2 with the indole ring of Trp479 and an exit from the pocket at approximately 6 ns. Interestingly, at 8 ns, the compound established interactions with amino acids of ACE2: Ala71, Lys74, Glu75, and Thr78. A relevant mutation in the Beta variant was found immediately next to the pocket studied. Specifically, in the E484K mutation, lysine replaces glutamic acid. Wang et al. [25] observed that the E484K mutation results in more favorable electrostatic interactions by significantly improving the binding affinity between the spike protein and the ACE receptor. In addition, this mutation leads to a rearrangement of the loop in which it is located, resulting in a tighter binding interface between the RBD and ACE2 and the formation of new hydrogen bonds. Interestingly, BCC compounds are located in this area, interacting not directly with the amino acid in question but with amino acids adjacent to it, suggesting they may behave as disruptors of S1AS complex interactions. For the Delta variant, the best results were obtained for the pockets located at the interface of the complex S1AS, that is, pockets 2 and 3, located in a nearly overlapping manner. The pyrimidine ring of BCC1 interacted and established a hydrogen bond with the NH of the amide group of Ser464 during the first 2 ns of the analysis. The benzene ring of the bicyclic system interacted with the imidazole ring of His448 for 5 ns from the beginning of the simulation. After 6 ns, the loop from Val461 to Leu481 underwent a conformational change leading to the exit of BCC from pocket 2, approaching pocket 1 for the remainder of the simulations. BCC2 briefly interacted with pocket 2, with occasional weak hydrophobic interactions with amino acids in the pocket. After approximately 2 ns, BCC2 exited the pocket and briefly interacted with Trp479 through π–π interactions between the quinolinium ring of the ligand and the imidazole ring of the amino acid, and then positioned itself more centrally at the interface between the spike protein and ACE2 for the remainder of the simulation. Molecular docking studies showed that the BCC3 ligand interacted with pocket 3 but not with pocket 2. The ligand briefly established hydrophobic interactions with the amino acids of ACE2 (Lys26 and Leu29), leaving pocket 3 after approximately 2 ns. After leaving the pocket, the ligand moved to a more central zone at the interface between the spike protein and ACE2, suggesting that this could be an area where these ligands are better accommodated. An interesting aspect of the Delta variant is that it loses the E484K mutation, which appeared in previous variants and expresses a new major mutation, T478K. Mahmood et al. and Cherian et al. showed how mutations in the Delta variant RBD enhance the interaction of the S1AS complex and its increased stabilization. BCC compounds interact with amino acids adjacent to those mutated in the Delta variant. This could lead to the occupancy of the interaction sites with ACE2, and, thus, the disruption of spike–ACE2 binding [26,27]. For the Gamma variant, pocket 4 showed the most interesting results. BCC1 remained in pocket 4 for the duration of the simulation. Initially, the molecule entered the pocket with its pyridine portion and then rotated after approximately 2 ns, establishing interactions with its bicyclic portion. The pyrimidine ring interacted with Asn440 to form H-bonds with the NH of the amino acid for the first 2 ns and then resumed the interaction at 8 ns until the end of the dynamics. Throughout the duration of the dynamics, the indole residue of Trp479 established intermittent π–π interactions with the bicyclic portion of the ligand, ensuring that it remained in the pocket for the duration of the simulation. From 3 to 6 ns, the pyrimidine ring was also involved in hydrogen interaction with an amino acid of ACE2 (Lys31). BCC2 remained in the pocket for only the first 4 ns and then moved around ACE2. In the first nanoseconds of the simulation, π–π interactions between the indole ring of Trp479 and the benzene ring of the bicyclic portion of the ligand and the pyrimidine ring of the ligand established a hydrogen bond with the NH of Asn440. After a few nanoseconds it moved away, into the receptor area of ACE2, establishing brief and weak hydrophobic interactions with the amino acids Lys68, Ala80, and Gln81. BCC3 maintained the same behavior. After 5 ns, the ligand left the pocket but did not establish interactions with the protein surroundings. Interactions established in the first 5 ns occurred between the indole ring of Trp479 and the benzene ring of the ligand. Moreover, for the Gamma variant, as for Beta, the E484K mutation was not located in the pocket and did not establish interactions with the ligands but in the surrounding pocket, allowing us to make the same point about the possibility that BCCs may be interposed in the area of interaction between spike protein and ACE2. Pocket 1 of the Omicron variant was the one that gave the best results. Specifically, BCC1 remained inside the pocket throughout the simulation because of hydrogen bonding between its pyrimidine ring and the indole ring of Trp479. This confirmed the docking results in which the ligand was inserted vertically into the pocket owing to its bicyclic system. BCC2 fitted into the pocket with its bicyclic system. Notably, after 8 ns, the quinolinium ring portion tended to leave the pocket; however, the pyrimidine moiety stays in the pocket thanks to hydrogen bond interactions with the oxygen of the carbamide group of Ala475 (up to 10 ns) and the NH of Asn477 (for the duration of dynamics). This variant expresses 15 mutations in the RBD of the spike protein. Rath et al. showed that, despite the high number of mutations, the RBD of the Omicron variant interaction with the spike protein is enhanced, and the S1AS complex is more stable. BCC compounds interpose themselves in the interaction zone between spike and ACE, probably resulting in an obstruction in the establishment of the interaction between the two proteins [28].

Moreover, the detailed energy values calculated by the various methods in the WaterSwap analysis are shown in Table 3. The results indicated that all of the ligand complexes showed good binding energies. The value under consideration is the consensus among the different methods for calculating the free energy of binding.

### 2.3. Quantum Mechanical Studies

MD simulations have shown that the ligand lies in the “pocket”, since the initial events. ONIOM calculations have indicated that the most effective interactions for these configurations have been due to weak attractive forces acting above and below the ligand, which have stabilized the complex. Conversely, when the ligand leaves the S1AS pocket, only one side of the ligand is exposed to the interaction, as depicted in Figure 10. Weak interactions with amino acid residues 75, 79, and 82 occur in all cases, but significant interactions with residues 476, 478, and 479 distinguish the most stable conformations, namely C2, C3, and C4. Compared to C1, where these residues are over 0.5 nm from the ligand, a stabilizing contribution between 5 and 10 kcal/mol is observed due to these residues, as shown in Table 4. A remarkable contribution of residues 478 and 479 to the interaction energy is also observed in C5, which is the most stable conformation among those selected after 8 ns (about 10 kcal · mol^−1^ lower). In general, interactions with residues 75 and 78 are present in all studied configurations. These findings confirm that Tyr478 and Trp479 play a significant role in stabilizing the ligand, while the carboxylic group on residue Glu75, whose interaction is almost always observed, could recognize the first event interaction through electrostatic interaction with the positively charged pyridinium group.

## 3. Discussion

To investigate the behavior of push–pull intercalating DNA compounds (BCCs) on the SARS-CoV-2 spike protein, the molecular modeling studies described above were performed. Pockets were searched for each variant and wild-type virus, and our results showed that mutations in amino acid sequences greatly influenced the search for likely active sites. One of the most interesting mutations in the Alpha variant is N501Y, where asparagine mutates to tyrosine at position 501, situated in the RBD receptor-binding domain. This mutation plays a role in enhancing binding with ACE2 [29,30]. The Gamma variant contains mutations, such as N501Y, E484K, and K417T in the RBD [31]. Comparing the Alpha variant with Gamma, we found two more pockets for the Gamma variant, which could be attributed to the different number of mutations between the two variants. Strangely, although there were no different mutations in the RBD between Gamma and Beta, the software found different numbers of pockets [32]. This could be attributable to the different amino acid mutations occurring throughout the spike protein, which could allow a different arrangement of the protein, translating into the software’s ability to find optimal pockets. Molecular dynamics simulations showed that compounds can interact with the S1AS complex, particularly by entering pockets at the interface between the two subunits. Specifically, pocket 3 for the Beta variant, pocket 4 for the Gamma variant, pockets 2 and 3 for the Delta variant, and pocket 1 for the Omicron variant. This is probably due to the ability of the ligands to enter this narrow region due to their planar conformation, interacting with the amino acids of the receptor domain of the spike protein, which extends from the amino acid Thr333 to Gly526 [20]. In addition, similar entering behaviors were observed for all compounds; BCCs entered the pocket by positioning themselves with their bicyclic portion, mostly establishing π–π interactions. BCC1, with its pyridine portion, established the most interactions and remained in various pockets for the duration of the dynamics or most nanoseconds of the simulation. In contrast, the BCC2 ligand, due to the presence of the quinolinium ring, is more rigid and less rotatable, which prevents it from establishing interactions that allow it to remain within the pocket for the duration of the dynamics. In fact, the ligand tends to leave the pocket after establishing brief initial interactions with some amino acids in the pocket and then moves away in search of more robust interactions. The BCC3 results showed that it maintained the least number of interactions, leading to the removal of the ligand from the pocket. This is plausible given that its structure contains an electron-poor imidazolium salt replaced by two methyl groups. These increase the steric bulk of a compound, decreasing its ability to establish interactions and increasing steric clashes. Furthermore, as mentioned above, these compounds are push–pull structures, π-conjugated systems characterized by a donor (D) and an acceptor (A) of electrons (D–π–A). It has been observed that the incorporation of heterocycles into these D–π–A systems gives this class of compound stability and conformational robustness, increased solubility, simplified synthesis, the possibility of eventual structural modification, and application in biological systems [33]. In the case of BCC compounds, an ethylene spacer separates quinolinium, pyridinium, or imidazolium salt (functional groups acting as acceptors) from a bicyclic 4-(pyrimidin-5-yl)-phenyl system (acting as an electron donor), and all three have iodide as a counterion. Previous photochemical studies of these compounds have shown that the quinolinium group is a better electron acceptor than the pyridinium group, while the imidazole group is the worst electron acceptor because of its π-deficient character [24]. Previous studies with molecules that share structural similarities with our compounds (planarity, condensed aromatic rings, and presence of heterocycles) further confirmed the good scores obtained with our FLAP docking [34]. Two molecules from a previous study were chosen to validate our screening results (methylene blue and DRI-C71041) [34]. These two molecules showed interaction scores similar to the BCCs. For example, we see that for pocket 1 of the wild-type, the Glob-Sum score is 1.392, and for pocket 3 of the Omicron variant, it is 1.647. The scores were almost superimposable with those obtained by the BCCs (Table 2).

## 4. Materials and Methods

### 4.1. In Silico Mutagnesis

The crystal structure of the S1 spike portion of SARS-CoV-2 bound to the ACE2 receptor was obtained from the protein data bank (PDB code: 6M0J resolution 2.45 Å) [35]. Flare 6.0 software (Cresset^®^, Litlington, Cambridgeshire, UK) [36] was used to modify the protein structure. In particular, the accessory parts of the spike protein and ACE2 receptor were removed from the PDB. For ease of writing, the complex will be reported as S1AS (spike 1-ACE subunit). With the “mutate to” function of Flare, the amino acids involved were mutated to create proteins of the different variants. proteins were then prepared for molecular dynamics analysis using the “protein prep’ Flare function. Both portions were relaxed with two cycles of short dynamic simulations (1 × 2 ns) using a water box created using the same program.

### 4.2. Binding Site Identification and Molecular Docking Studies

All virtual screening studies were generated using FLAP 2.2.2 software (Molecular Discovery Ltd., Borehamwood, UK; www.moldiscovery.com (accessed on 13 January 2023)) [37]. FLAP describes small molecules and protein binding sites (called pockets) in terms of four-point pharmacophoric fingerprints extracted from molecular interaction fields (MIFs) calculated by GRID [38]. First, the interaction cavities (pockets) of the crystallographic protein were calculated using FLAP. The pockets were identified both automatically and manually. The pockets are identified automatically by the software with the “search by pocket” function, while the “search by residue” function allows the operator to select the amino acid residues of interest to obtain a pocket that the software is not able to obtain automatically. Moreover, FLAP software was used in the “structure-based’’ mode (SBVS); this function has the purpose of generating all possible binding poses of a ligand within a pocket (binding site). This process is based on finding similarities between the GRID fields of the ligand and the binding site [38,39]. GRID MIFs were generated using four molecular probes: H (shape and steric effects), DRY (hydrophobic interactions), N1 (H-bond donor), and O (H-bond acceptor) interactions. SBVS function overlaps the grids and scores it; values were extrapolated for each compound, resulting in a total of 19 different scores, referring to the single probes and in combinations between each probe. In addition, the function returns three other important scores for evaluating interactions: Glob-Sum, Glob-Prod, and Distance. The first two values refer to the summation and production of interactions, respectively. The Distance score represents the overall similarity derived from a combination of the degree of overlap between the individual probes (H, DRY, O, and N1) of the MIFs, calculated for each candidate ligand and binding site.

The SBVS function was performed using the X-ray crystal structures of the S1-CTD portion of the spike protein. Glob-Sum was used as a reference score to quantify the degree of interaction between the ligand and the protein active site because it provides more indicative and reliable data than Glob-Prod for the reasons mentioned above. The other scores serve to further rationalize each screening. The N1 (hydrogen bond donor) value for each pocket of each variant is 0. The wild-type virus, as well as the mutated forms, Alpha, Beta, Gamma, Delta, and Omicron variants, have been studied using this method. Under the same conditions, screening was performed on known compounds (methylene blue and DRI-C71041) to validate and compare our results and interaction scores [34].

### 4.3. Molecular Dynamics Studies

The structures of molecules BCC1, BCC2, and BCC3 were retrieved from the aforementioned molecular docking studies, and their energy was minimized with a “Flare preparation ligand.” The ligand poses found to be the best in molecular docking studies were used for dynamics studies. Short molecular dynamics runs were also performed at a constant temperature with a minimization of the atoms in the binding site. This procedure was used to improve the stability of these complexes. The most plausible poses with the best binding energy scores derived from docking studies were chosen for molecular dynamics analysis. A tLeap water box (TIP3PBOX) was produced. The water–protein system (grid dimensions: 68 × 54 × 136 Å) was minimized. The forcefield used for proteins was AMBER FF14SB, and for ligands, AMBER GAFF2. Five dynamic simulations were performed over 50 ns. Dynamic analysis at 50 ns and root-mean-square deviation (RMSD) evaluation showed no significant changes. Dynamic files were visualized and analyzed using Flare 6.0 software. Furthermore, to calculate the binding free energies of the ligand-protein complexes investigated, the WaterSwap absolute binding free energy method was applied using a function in the Flare software. The binding free energy was calculated using the Bennett method, thermodynamic integration (TI), and free energy perturbation (FEP). The final values of binding free energy were obtained by calculating the arithmetic mean of the energies determined by previous methods.

### 4.4. Quantum Mechanical Studies

The initial geometries of the BCC-1-S1AS complexes were obtained from molecular dynamics simulations, in particular, four different geometries were selected within 6 ns of MD simulations (henceforth named C1, C2, C3, and C4) and four geometries were chosen after 8 ns of simulation time (C5, C6, C7, and C8). The complexes were fully optimized without constraints at the ONIOM (CAM-B3LYP6-311+G(d,p):AMBER) level of theory. According to the ONIOM procedure [40], the molecular system was separated into two different layers, which were simulated with different model chemistries. The DFT approach was applied to describe the high layer represented by the ligand BCC-1 together with all atoms of the S1AS structure within 0.5 nm of the ligand, whereas the AMBER forcefield was employed to simulate the low layer (Figure 11).

According to this approach, the energy EONIOM is defined as:EONIOM = Elow (R) + Ehigh (SM) − Elow (SM)
where Elow (R) is the energy of the real system, obtained with the AMBER forcefield, Ehigh (SM) and Elow (SM) are the energies of the small model calculated at DFT and AMBER levels of theory, respectively. As mentioned above, to treat the closer interactions at the DFT level, the atoms around the ligand were included in the high layer; following this approach X-Y (where X = C and Y = C, N or O) sigma bonds were cut employing hydrogen link atoms, hence all dangling bonds in the small model were capped with H atoms. The polarizable continuum model (PCM/X) was used to describe the solvation effects [41]. Finally, all of the calculations were run using the Gaussian 16 software package [42].

The optimized molecular geometries of the studied complexes are reported in Figure 10. Table 4 shows the electronic interaction energy values, ΔE, calculated as follows:E_int_ = E_BCC-1-S1AS_ − (E_BCC-1_ + E_S1AS_)

## 5. Conclusions

In this study, the series of BCC compounds that behave as DNA intercalants were found to be able to interact with the RBD of the spike protein of SARS-CoV-2. This is important because it shows that they could lead to disruption of the spike–ACE2 interaction and prevent viral recognition and entry into cells. These lead compounds will be investigated in future studies to understand which structural modifications are most appropriate for improving selectivity for the spike protein. From the molecular modeling data supported by quantum mechanical studies, it is clear that some interaction zones are prioritized over others. This is supported by evidence that FLAP software found several pockets for each variant; however, in molecular dynamics studies, those that performed best were those that were at the interface of the SA1S complex. In particular, it is important to note that in the different viral variants, compounds occupied zones and interacted with residues of the spike protein that are considered critical in the recognition of the ACE2 receptor by spike.

## Figures and Tables

**Figure 1 molecules-28-03908-f001:**
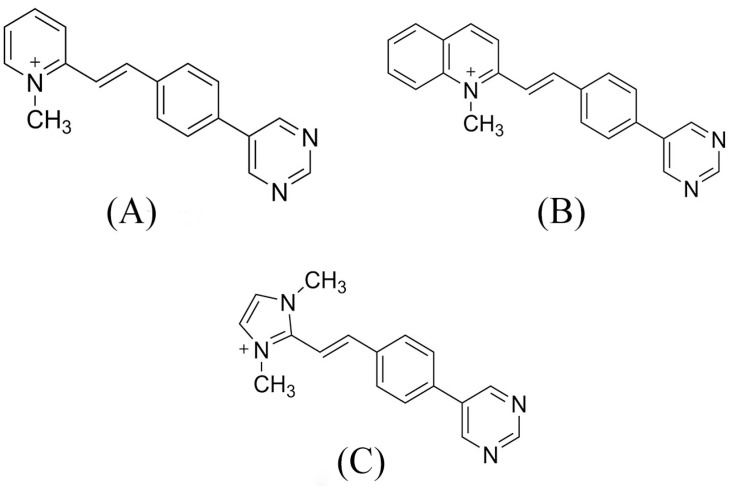
Push–pull compounds structures: (**A**) (*E*)-1-methyl-2-4-(pyrimidin-5-yl)styryl) pyridin-1-ium (BCC1), (**B**) (*E*)-1-methyl-2-4-(pyrimidin-5-yl)styryl) quinolin-1-ium (BCC2), and (**C**) (*E*)-1,3-dimethyl-2-4-(pyrimidin-5-yl)styryl)-1-h-imidazol-3-ium (BCC3).

**Figure 2 molecules-28-03908-f002:**
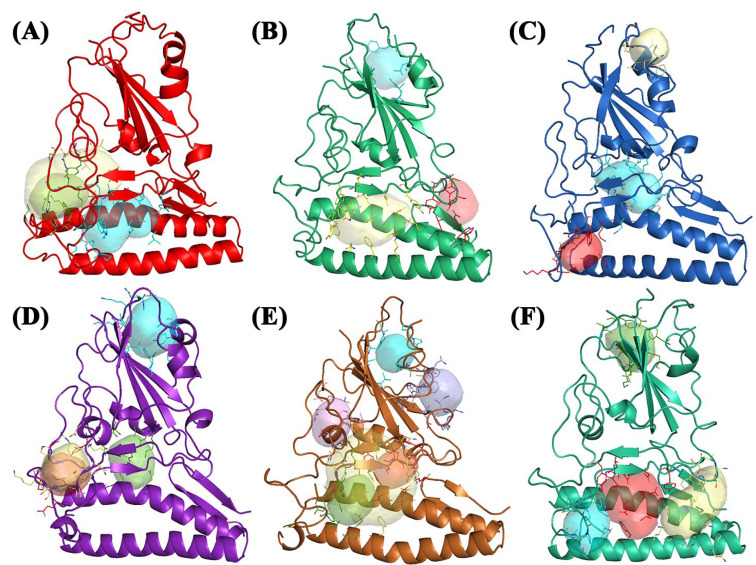
(**A**) Wild-type: pocket 1 (blue), pocket 2 (green), and pocket 2.5 (yellow); (**B**) Alpha: pocket 1 (blue), pocket 2 (yellow), and pocket 3 (red); (**C**) Beta: pocket 1 (blue), pocket 2 (yellow), and pocket 3 (red); (**D**) Delta: pocket 1 (blue), pocket 2 (yellow), pocket 3 (red), and pocket 4 (green); (**E**) Gamma: pocket 1 (blue), pocket 2 (violet), pocket 3 (red), pocket 3.5 (yellow), pocket 4 (green), and pocket 5 (dark blue); and (**F**) Omicron: pocket 1 (blue), pocket 2 (yellow), pocket 3 (red), and pocket 4 (green). Water and the accessory parts of the spike protein and ACE2 receptor were omitted for clarity.

**Figure 3 molecules-28-03908-f003:**
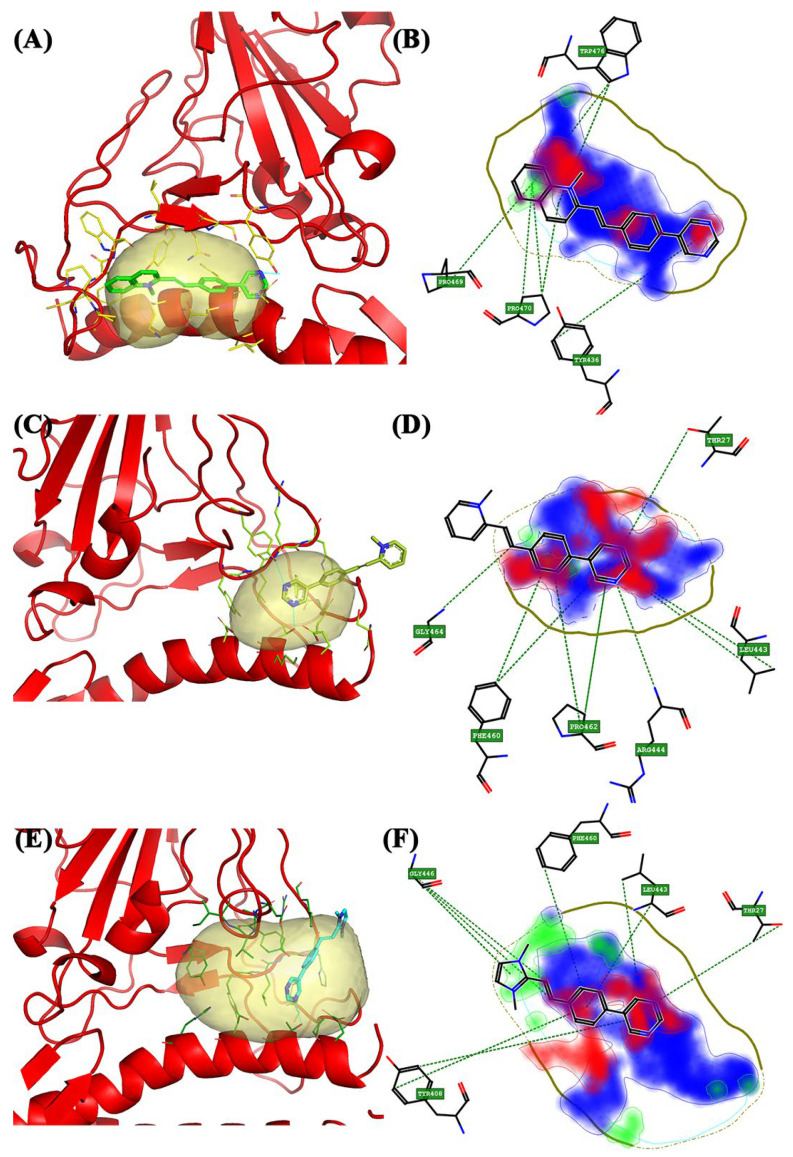
The 3D and 2D docking wild-type: (**A**) 3D pocket 1 with BCC2; (**B**) 2D pocket 1 with BCC2; (**C**) 3D pocket 2 with BCC1; (**D**) 2D pocket 2 with BCC1; (**E**) 3D pocket 2.5 with BCC3; and (**F**) 2D pocket 2.5 with BCC3. Water and the accessory parts of the spike protein and ACE2 receptor were omitted for clarity. The text in the green square of the image describes the amino acids involved in interaction with the ligand.

**Figure 4 molecules-28-03908-f004:**
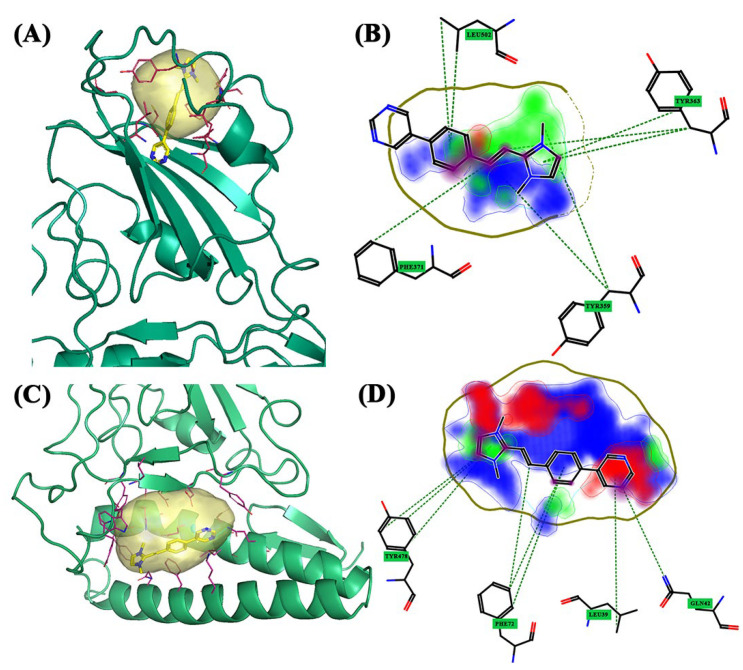
The 3D and 2D docking Alpha variant: (**A**) 3D pocket 1 with BCC3; (**B**) 2D pocket 1 with BCC3; (**C**) 3D pocket 2 with BCC3; and (**D**) 2D pocket 2 with BCC3. Water and the accessory parts of the spike protein and ACE2 receptor were omitted for clarity. The text in the green square of the image describes the amino acids involved in interaction with the ligand.

**Figure 5 molecules-28-03908-f005:**
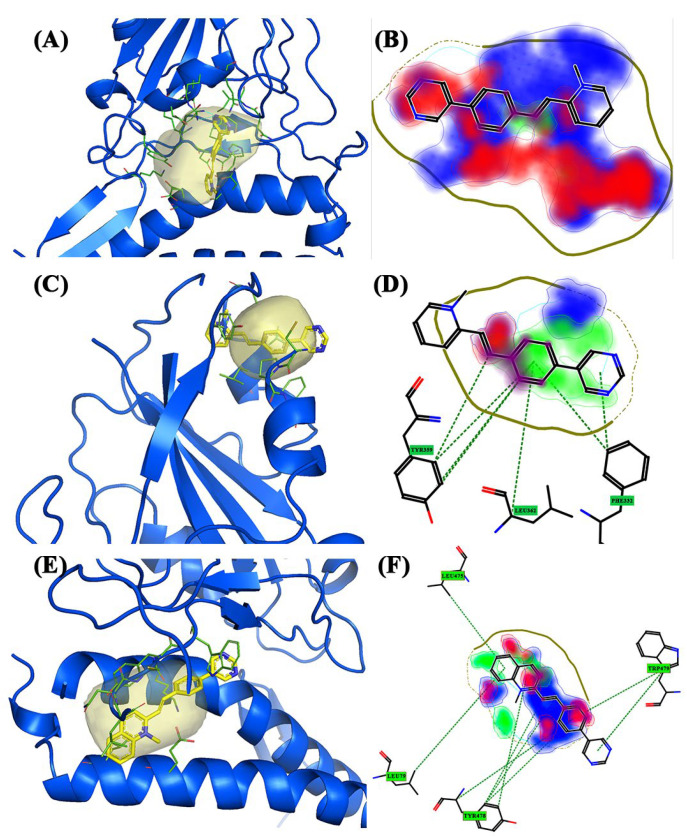
The 3D and 2D docking Beta variant: (**A**) 3D pocket 1 with BCC1; (**B**) 2D pocket 1 with BCC1; (**C**) 3D pocket 2 with BCC1; (**D**) 2D pocket 2 with BCC1; (**E**) 3D pocket 3 with BCC2; and (**F**) 2D pocket 3 with BCC2. Water and the accessory parts of the spike protein and ACE2 receptor were omitted for clarity. The text in the green square of the image describes the amino acids involved in interaction with the ligand.

**Figure 6 molecules-28-03908-f006:**
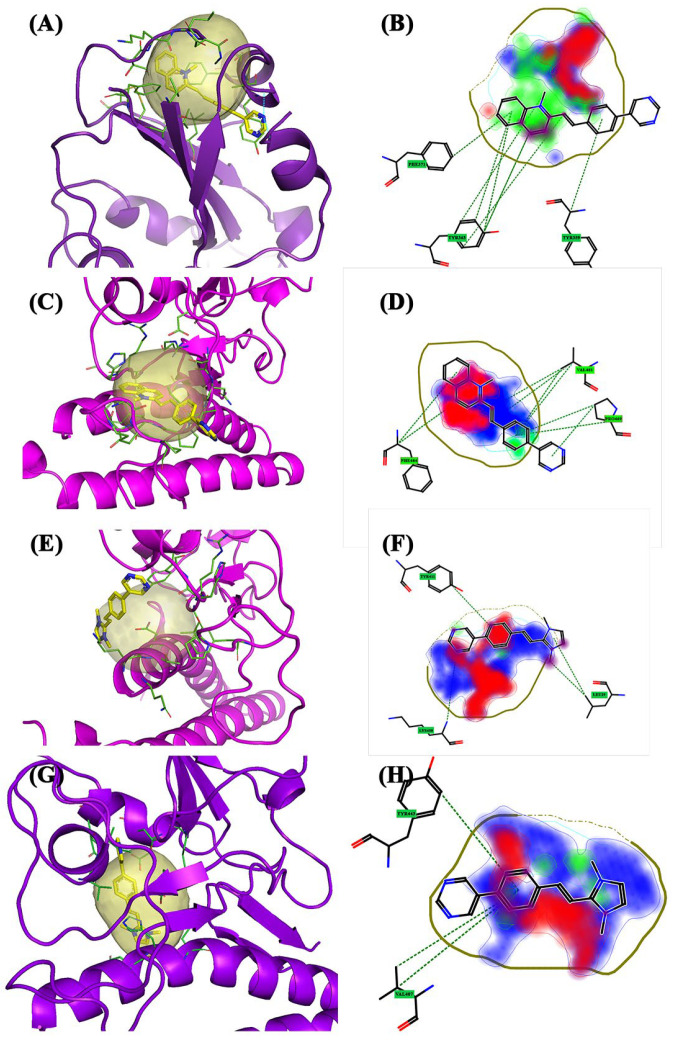
The 3D and 2D docking Delta variant: (**A**) 3D pocket 1 with BCC2; (**B**) 2D pocket 1 with BCC2; (**C**) 3D pocket 2 with BCC2; (**D**) 2D pocket 2 with BCC2; (**E**) 3D pocket 3 with BCC3; (**F**) 2D pocket 3 with BCC3; (**G**) 3D pocket 4 with BCC3; and (**H**) 2D pocket 4 with BCC3. Water and the accessory parts of the spike protein and ACE2 receptor were omitted for clarity. The text in the green square of the image describes the amino acids involved in interaction with the ligand.

**Figure 7 molecules-28-03908-f007:**
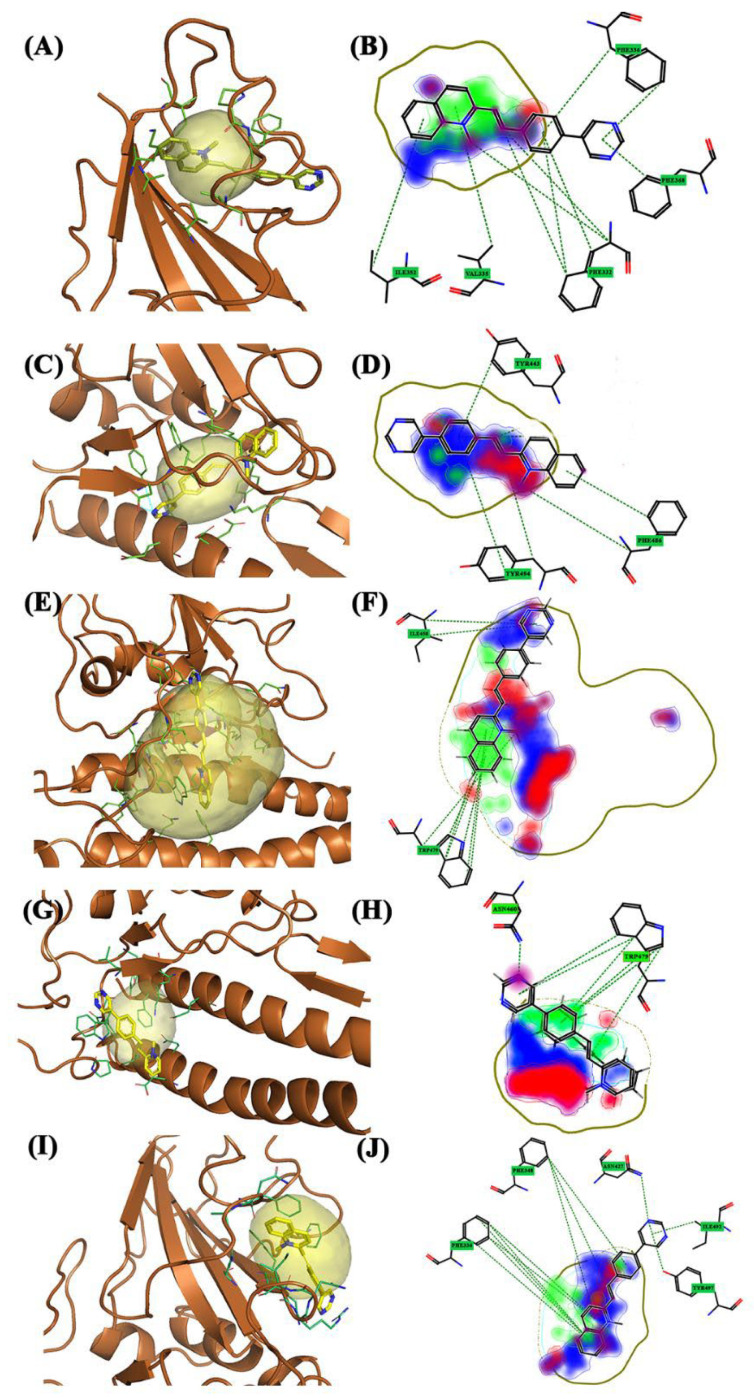
The 3D and 2D docking Gamma variant: (**A**) 3D pocket 1 with BCC2; (**B**) 2D pocket 1 with BCC2; (**C**) 3D pocket 3 with BCC2; (**D**) 2D pocket 3 with BCC2; (**E**) 3D pocket 3.5 with BCC2; (**F**) 2D pocket 3.5 with BCC2; (**G**) 3D pocket 4 with BCC1; (**H**) 2D pocket 4 with BCC1; (**I**) 3D pocket 5 with BCC2; and (**J**) 2D pocket 5 with BCC2. Water and the accessory parts of the spike protein and ACE2 receptor were omitted for clarity. The text in the green square of the image describes the amino acids involved in interaction with the ligand.

**Figure 9 molecules-28-03908-f009:**
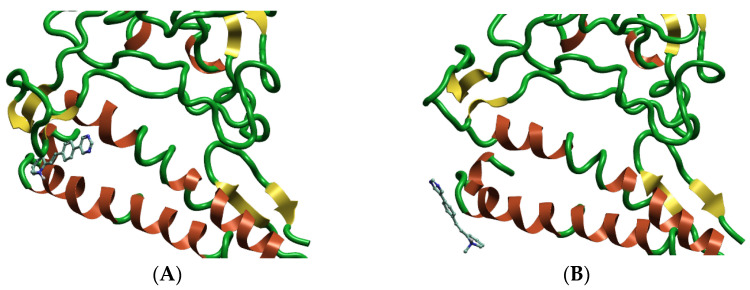
MD simulations of BCC1 in pocket 3 of the Beta variant: (**A**) Conformation of the loop from amino acid Val461 to Leu481; and (**B**) conformational change in the loop from amino acid Val461 to Leu481 leads to the exit of the ligand from the pocket. Water and the accessory parts of the spike protein and ACE2 were omitted for clarity.

**Figure 10 molecules-28-03908-f010:**
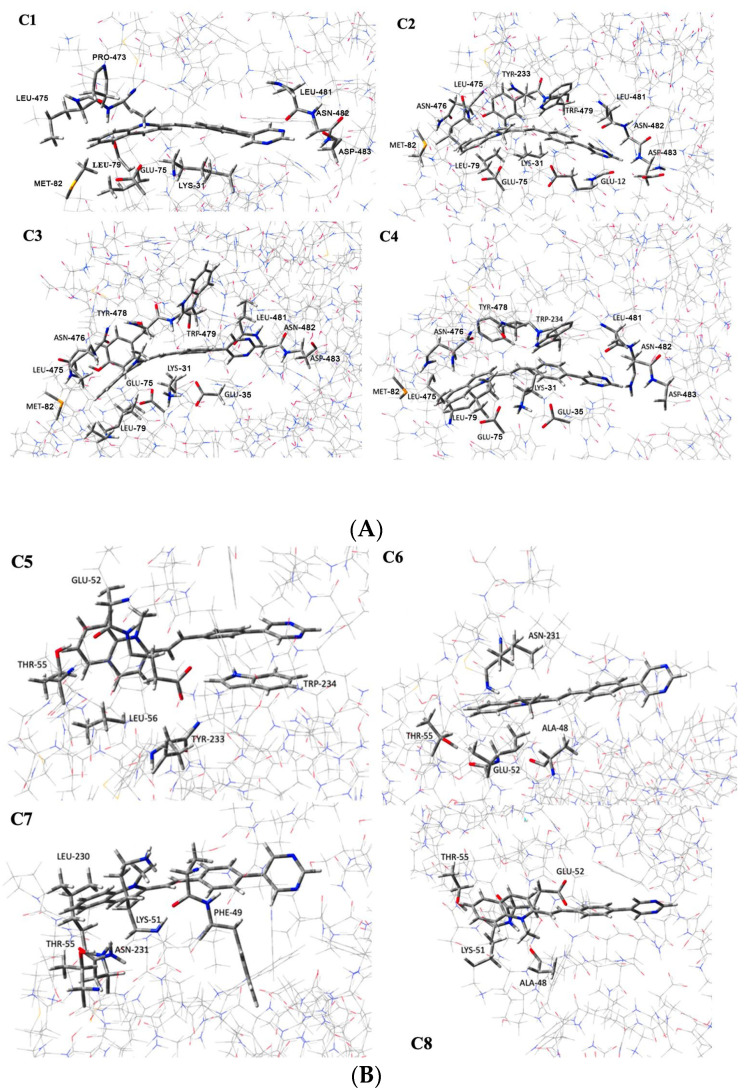
(**A**) ONIOM-optimized geometries of C1–C4. C1: geometry of the BCC1-S1AS complex and interactions in conformation 1; C2: geometry of the BCC1-S1AS complex and interactions in conformation 2; C3: geometry of the BCC1-S1AS complex and interactions in conformation 3; C4: geometry of the BCC1-S1AS complex and interactions in conformation 4. (**B**) ONIOM-optimized geometries of C5–C8. C5: geometry of the BCC1-S1AS complex and interactions in conformation 5; C6: geometry of the BCC1-S1AS complex and interactions in conformation 6; C7: geometry of the BCC1-S1AS complex and interactions in conformation 7; C8: geometry of the BCC1-S1AS complex and interactions in conformation 8.

**Figure 11 molecules-28-03908-f011:**
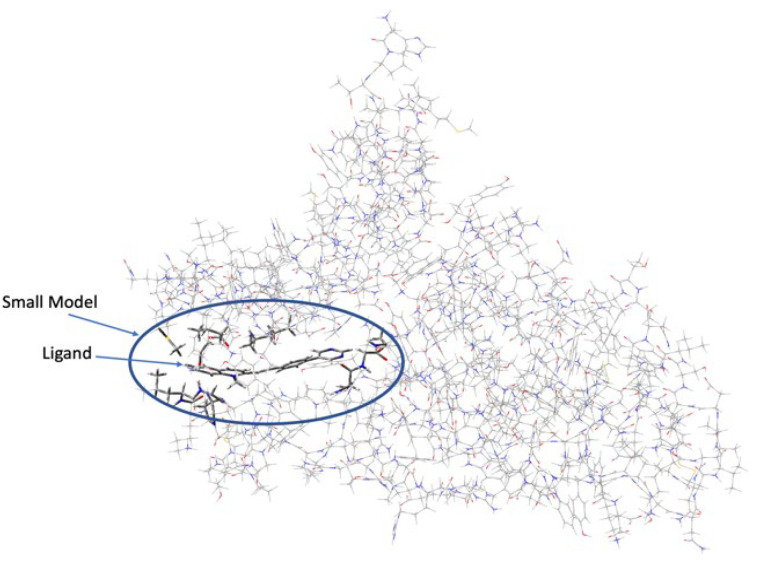
ONIOM model of the BCC-1-S1AS complex. The high layer is represented by the tube model, and in the wireframe, the low layer.

**Table 3 molecules-28-03908-t003:** Binding free energies of the BCC–S1AS complexes calculated by Bennet, TI, and FEP methods and relative consensus average.

Variant	Pocket	Ligand	Bennet (kcal/mol)	FEP (kcal/mol)	TI (kcal/mol)	Consensus (kcal/mol)
Beta	Pocket 3	BCC1	−19.84	−18.68	−17.97	−19.05 ± 1.87
	Pocket 3	BCC2	−19.45	−20.85	−20.45	−20.03 ± 1.40
Delta	Pocket 2	BCC1	−24.58	−22.22	−21.93	−23.31 ± 2.64
Gamma	Pocket 4	BCC1	−23.91	−19.63	−20.20	−21.94 ± 4.29
Omicron	Pocket 1	BCC1	−23.99	−20.19	−20.27	−22.11 ± 3.95

**Table 4 molecules-28-03908-t004:** Interaction energies ΔE of the BCC-1-S1AS model systems and amino acidic residues within a distance of 0.5 nm.

Configuration	ΔE (kcal · mol^−1^)	Interacting Residues
C1	−45.31	Glu35, Glu75, Leu79, Met82, Pro473, Lys474, Leu475, Leu481, Asn482, and Asp483
C2	−55.75	Glu35, Glu75, Leu79, Met82, Leu475, Asn476, Tyr478, Trp479, Leu481, and Asn482, Asp483
C3	−51.04	Glu35, Glu75, Leu79, Met82, Leu475, Asn476, Tyr478, Trp479, Leu481, and Asn482, Asp483
C4	−53.72	Glu35, Glu75, Leu79, Met82, Leu475, Asn476, Tyr478, Trp479, Leu481, Asn482, and Asp483
C5	−35.10	Glu75, Thr78, Leu79, Tyr478, and Trp479
C6	−20.14	Glu75, Ala71, Thr78, and Asn476
C7	−26.26	Phe72, Lys74, Thr78, Leu475, and Asn476
C8	−20.29	Ala71, Lys74, Glu75, and Thr78

## Data Availability

Not applicable.

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
