# Peer review of "Different In Silico Approaches Using Heterocyclic Derivatives against the Binding between Different Lineages of SARS-CoV-2 and ACE2"

_molecules, 2023, doi:10.3390/molecules28093908_

Round 1

Reviewer 1 Report

The COVID-19 pandemic is one of the leading causes of death worldwide as new SARS-CoV-2 variants continue to emerge. These emerging variants primarily involve mutations in the Spike (S) protein which is responsible for entry of the SRAS-CoV-2 virus into host cells. Understanding the molecular structure of the S protein and its interaction with its receptor, angiotensin-converting enzyme 2 (ACE2), provides a fundamental understanding of viral biology as well as therapeutic strategies, including small-molecule therapeutics. Sipala et al. used different in silico approaches to discover that heterocyclic derivatives could be potential inhibitors of S protein by preventing ACE2 binding. This study is certain useful in this field; however, I have several criticism that preclude this manuscript to be published in the current form.

Comments

1.    This paper lacks proper acknowledge of existing efforts of small-molecule discovery targeting S proteins including PMID: 36844485, 34349236 etc… Therefore, it is unclear to me what were the prior efforts in discovering small molecule therapeutics targeting S protein and what is the novelty of this current research.

2.    This research appears to have been conducted without proper controls. For example, we have never seen the author verify binding pocket identification and molecular docking with known her S-protein binders.

3.    What previous work is the author referring to on lines 81-83? What is the biological activity?

4.    Line 86-87, as these compounds have been shown to be DNA intercalators, does this raise potential safety concerns as they may preferably interact with DNA and interfere with DNA biochemical process? Do these properties make it a better candidate to target S protein?

5.    For Figure 2, could the author indicate in a separate penal the regions of RBD that interactions with ACE2? It is important to highlight whether these identified binding pockets are binding pockets for ACE2 or allosteric pockets.

6.    The 2D pocket illustrations need to be changed. The electron maps (red, blue, and green color) blur the visualization of interactions between protein and small molecules. The author can have a separate figure penal to show the electron map of the compound.

7.    Since there are no differences in mutations in the RBD between gamma and beta variants, did the author compare the structures between them to understand why the binding pockets were identified differently? To me, it is a little surprising that the binding pockets are quite different between them.

8.    What force field did the authors use in their MD simulation? Have the authors tried other force fields and shown that the binding of these small molecules is robust? The author also mentioned the simulations did not show relevant variations up to 50 ns. What is the metric for measuring variations?

Author Response

Response to Reviewer 1 Comments

Point 1: This paper lacks proper acknowledge of existing efforts of small-molecule discovery targeting S proteins including PMID: 36844485, 34349236 etc… Therefore, it is unclear to me what were the prior efforts in discovering small molecule therapeutics targeting S protein and what is the novelty of this current research.

Response 1: Citations to the article have been added to line 78. The purpose and results of this study allow us to further characterize the interaction between the different mutated variants of the SARS-CoV-2 spike protein and the use of small molecules to prevent the interaction of the spike protein with human ACE2.

Point 2: This research appears to have been conducted without proper controls. For example, we have never seen the author verify binding pocket identification and molecular docking with known her S-protein binders. 

Response 2: Thank you for this comment. Validation checks were added in line 750 to confirm the FLAP scores obtained from our screening. Two molecules (methylene blue and DRI-C71041) were chosen from the study by Bojadzic et al. (2021), which presented structural similarities with our molecules (planarity, condensed aromatic rings, and the presence of heterocycles) and good interactions with the Spike Protein.

Point 3: What previous work is the author referring to on lines 81-83? What is the biological activity?

Response 3: We have referred to the work in Barresi, V.; Bonaccorso, C.; Consiglio, G.; Goracci, L.; Musso, N.; Musumarra, G.; Satriano, C.; Fortuna, C.G. Modeling, design, and synthesis of new heteroaryl ethylenes active against MCF-7 breast cancer cell line Molecular bioSystems 2013, 9, 2426-2429, DOI 10.1039/c3mb70151d. This study demonstrated the antitumor activities of several compounds against MCF-7 breast cancer cells.

Point 4: Line 86-87, as these compounds have been shown to be DNA intercalators, does this raise potential safety concerns as they may preferably interact with DNA and interfere with DNA biochemical process? Do these properties make it a better candidate to target S protein? 

Response 4: This research is intended to be basic research aimed at identifying lead compounds for the goal proposed in this paper. The compounds tested exhibit intercalation activity in DNA, as seen by Barresi et al (2013) and Mazzoli et al (2011), but as lead compounds, they will undergo further studies aimed at identifying the best substituents and most appropriate structural modifications to make these compounds selective toward the target proposed in this study.

Point 5:  For Figure 2, could the author indicate in a separate penal the regions of RBD that interactions with ACE2? It is important to highlight whether these identified binding pockets are binding pockets for ACE2 or allosteric pockets.

Response 5: Thank you for this comment. The text and images have been implemented with a representative table of all amino acids involved in the pockets, also highlighting which pockets involve ACE2 amino acids.

Point 6: The 2D pocket illustrations need to be changed. The electron maps (red, blue, and green color) blur the visualization of interactions between protein and small molecules. The author can have a separate figure penal to show the electron map of the compound. 

Response 6: The 2D representations obtained using FLAP lose their meaning if they are deprived of colored interaction areas. The green, red, and blue areas are important for understanding how compounds interact with the pockets.

Point 7: Since there are no differences in mutations in the RBD between gamma and beta variants, did the author compare the structures between them to understand why the binding pockets were identified differently? To me, it is a little surprising that the binding pockets are quite different between them. 

Response 7: In our discussion, we have commented on this observation. Although there were no different mutations in the RBD between Gamma and Beta, the software found different numbers of pockets. While these two variants have the same mutations in the RBD, they do not have identical mutations in the rest of the spike protein. This could allow for a different arrangement of the protein, translating the software's different abilities to find different optimal pockets between the two variants.

Point 8: What force field did the authors use in their MD simulation? Have the authors tried other force fields and shown that the binding of these small molecules is robust? The author also mentioned the simulations did not show relevant variations up to 50 ns. What is the metric for measuring variations?

Response 8: Proteins were subjected to minimization using the Amber14 FF14SB force field. The ligands were constructed and minimized by applying a GAFF2 force field. Evaluations were performed in terms of RMSD root mean square deviation, which allowed us to analyze the molecular dynamics trajectories and observe that in the average of the 5 replicates performed, the protein-ligand complex was stabilized for all 50 ns. These information has been added in the materials and methods line 824.

Reviewer 2 Report

The article is devoted to an important problem: the search for drugs for COVID-19 infection. 

Considerable work has been done, but the presentation of the results seems to me

rather cumbersome and confusing. 

There are many abbreviations in the text, but not all of them are deciphered. It is necessary to highlight a separate

part of the article, where it is necessary to give a transcript and explanation of all accepted abbreviations.

Most of the article is devoted to the presentation of the docking regulations (2.1. Binding site identification and Molecular Docking Studies).

There are many figures in this part, but there is not a single table. It seems to me that this part should be further structured and provided with tables.

It remains unclear to me why, after finding the binding sites, the authors switch to quantum calculations at a fixed distance of the ligand receptor (5 angstroms). I don't understand the meaning of the formula: ΔE = EBCC-1-S1AS – EBCC-1 – ES1AS. This formula is at least incorrect from the point of view of dimension. (kcal/mole (right) is not dimensionless value as 1 (left) is)

The second step after docking should be to study the binding energy and the lifetime of the complex in the bound state. These data can be obtained based on molecular dynamics data using one of the standard techniques (MM/PBSA MM/GBBSA).

Quantum calculations are used to optimize valence bonds and usually precede calculations of docking and molecular dynamics. 

The need for quantum calculations in this article should be justified separately. Similarly, the choice of a fixed distance should be justified.

The lack of estimates of binding energies despite the fact that the molecular dynamics trajectories have already been obtained remains unclear. The article looks unfinished in this regard.

Author Response

Response to Reviewer 2 Comments

Point 1: The article is devoted to an important problem: the search for drugs for COVID-19 infection.

Considerable work has been done, but the presentation of the results seems to me rather cumbersome and confusing. There are many abbreviations in the text, but not all of them are deciphered. It is necessary to highlight a separate part of the article, where it is necessary to give a transcript and explanation of all accepted abbreviations.

Response 1: Thank you for your comment. We have added a paragraph with abbreviations in the text (line 891).

Point 2: Most of the article is devoted to the presentation of the docking regulations (2.1. Binding site identification and Molecular Docking Studies). There are many figures in this part, but there is not a single table. It seems to me that this part should be further structured and provided with tables.

Response 2: The text and images have been implemented with a representative table of all amino acids involved in the pockets (line 538), and a table showing all the docking results (541).

Point 3: It remains unclear to me why, after finding the binding sites, the authors switch to quantum calculations at a fixed distance of the ligand receptor (5 angstroms). I don't understand the meaning of the formula: ΔE = EBCC-1-S1AS – EBCC-1 – ES1AS. This formula is at least incorrect from the point of view of dimension. (kcal/mole (right) is not dimensionless value as 1 (left) is).

Response 3: ONIOM approach was used to estimate the weak interactions (VdW, H-bonds,…) between ligand BCC-1 and substrate S1AS, by using a finer approach, (the QM method) compared to the classical description (see ref. 40 for details). In particular the ligands and all the atoms of S1AS closer than a cut off value, 5 Angstrom, were treated with the quantum mechanical method.

As to the following equation:

ΔE = EBCC-1-S1AS – EBCC-1 – ES1AS

we explicitly refer to ΔE as “electronic interaction energy” which is measured in kcal/mol, it represents the difference between the ligand-substrate system energy E(BCC-1-S1AS) and the energies of the substrate (ES1AS) and ligand (EBCC-1) . For clarity we have now reported the formula as:

Eint = EBCC-1-S1AS – (EBCC-1 + ES1AS)

Point 4: The second step after docking should be to study the binding energy and the lifetime of the complex in the bound state. These data can be obtained based on molecular dynamics data using one of the standard techniques (MM/PBSA MM/GBBSA).

Response 4: The binding free energies of the ligand-protein complexes were studied by applying the WaterSwap absolute binding free energy method using the Flare software function, which is based on the exchange between the protein-bound ligand and an equivalent volume of water molecules. The methods used to calculate the binding free energy were Bennett's method, thermodynamic integration (TI), and perturbation free energy (FEP). Finally, the arithmetic mean of the energies determined by these methods was considered. The method description and results are included in lines 655 and 674 (Table 3).

Point 5: Quantum calculations are used to optimize valence bonds and usually precede calculations of docking and molecular dynamics. The need for quantum calculations in this article should be justified separately. Similarly, the choice of a fixed distance should be justified.

Response 5: As explained in response 3, the ONIOM quantomechanical method was used to further investigate the behavior of the BCC-1 ligand and the S1AS substrate by estimating the weak interactions of the complex using a more accurate method.

Point 6: The lack of estimates of binding energies despite the fact that the molecular dynamics trajectories have already been obtained remains unclear. The article looks unfinished in this regard.

Response 6: As explained in response 4, the binding free energies of the ligand-protein complexes were studied by applying Flare's WaterSwap method, and the results are reported in Table 3 to improve the completeness of our work.

Reviewer 3 Report

The study has been done comprehensively together with the well-written manuscript. Here is my only comment:

The authors mentioned that the series of BCC compounds behave as DNA intercalants. As the DNA intercalants can cause DNA damage and even mutation, the authors should justify why these compounds should be considered to be used against SARS-CoV-2 infection given that current available options such as mRNA vaccines and neutralizing antibodies do not has such risky outcomes.

Author Response

Response to Reviewer 3 Comments

Point 1: The study has been done comprehensively together with the well-written manuscript. Here is my only comment:

The authors mentioned that the series of BCC compounds behave as DNA intercalants. As the DNA intercalants can cause DNA damage and even mutation, the authors should justify why these compounds should be considered to be used against SARS-CoV-2 infection given that current available options such as mRNA vaccines and neutralizing antibodies do not has such risky outcomes.

Response 1: From the in silico studies presented in this article, we conclude that the BCC series of heterocyclic compounds interact with the spike protein as disruptors of ACE2 receptor binding. The compounds tested show intercalation activity in DNA, as shown by previous studies by Barresi et al (2013) and Mazzoli et al (2011). This research is intended as basic research aimed at identifying lead compounds for the goal proposed in this paper. The results identified the studied compounds as lead compounds, and they will undergo further studies aimed at identifying the best substituents and the most congenial structural modifications to make these compounds selective toward the target proposed in this study.

Round 2

Reviewer 1 Report

The authors addressed all my previous comments and the manuscript now is acceptable with minor typo checks.

Reviewer 2 Report

It remains unclear to me why, instead of the well-known and standard method of evaluating binding, which is MMGBSA method The authors use a method that combines the quantum mechanical and molecular mechanical approach, which is adapted to the calculation program chosen by the authors. In my opinion, this is a bad practice, however, I believe that criticism of this approach should be left to the readers. Therefore, the article should be published in this form, given the value of the subject of research itself.